# Zero-shot Object Detection with a Text and Image Contrastive Model

## Abstract

We introduce *DUCE*, a generalizeable zero-shot object detector, and *BCC*, a novel method of bounding box consolidation for models where traditional non-maximum suppression is insufficient. *DUCE* leverages the zero-shot performance of CLIP (Radford et al. (2021)) in combination with a region proposal network (Ren et al. (2015)) to achieve state of the art results in generalized zero-shot object detection with minimal training. This approach introduces a new challenge in that *DUCE* is able to label portions of an image with very high confidence, leading to numerous high confidence bounding boxes around an object of interest. In these scenarios, traditional forms of non-maximum suppression fail to reduce the number of bounding boxes. We introduce *BCC* as a new approach to bounding box suppression, that allows us to successfully navigate this challenge. *DUCE* and *BCC* are able to achieve competitive results to other state of the art models for all classes, agnostic of whether or not the RPN was trained on those classes. Our proposed model and new method bounding-box consolidation represents a novel approach to the zero-shot object detection task.

## 1 Introduction

While modern object detection models are able to perform at superhuman levels on objects for which they are trained, recent high-profile failures of these systems, such as those caused by self-driving car accidents (McFarland (2020)), shows that a new approach is required which will allow object detection models to better detect objects and situations for which they have not been trained. While detectors such as YOLO (Redmon et al. (2015)) and FRCNN (Ren et al. (2015)) require training end to end on large datasets of hand labeled images, we seek to approach object detection in a different way.

In this paper, we introduce Detection of Unknown objects through CLIP Extraction (*DUCE*), an attempt to address the difficulty of generating a general zero-shot object detection model without requiring the collection and curation of large data sets as well as avoid the vast training times required for current state of the art object detection methods by proposing a new zero-shot object detection algorithm.

By combining the region proposal methods from FRCNN with the text-to-image classification of properties from CLIP, we are able to provide a zero-shot object detector capable of classifying objects on which it was never trained on with a recall of 77%. This paper presents two contributions. First, we present a zero-shot object identification model: *DUCE*. Second, we present a novel method of bounding box consolidation and suppression, that works especially well with pre-trained image embeddings.

## 2 Background

Object detection and image classification are two areas of computer vision that have gained research interest and popularity in recent years. Single shot models such as You Only Look Once (YOLO) (Redmon et al. (2015)) and two-stage approaches such as Faster R-CNN (Ren et al. (2015)), are some of the most popular approaches in object detection today. More recent techniques, such as the one introduced by Chen et al. (2021) view object detection in a similar way to existing language models, where the output from the detector

are the bounding box locations when asked the 'question' that is an image. Each of these methods require large amounts of training and exposure to multiple examples of new classes to properly and accurately learn to detect and label those objects (Dong et al. (2019)). Additionally, these methods are unable to detect objects outside of their initial training. Lack of labeled training data is a major impediment to most advanced machine learning, including computer vision projects (Mahony et al. (2019)). Gathering this training data is time consuming, expensive, and prone to error. Recent research (Northcutt et al. (2021)) indicates that error rates range from 0.15% to 10.12% in classification datasets. Object detection datasets introduce yet another source of error due to the addition of bounding box annotations. ImageNet (Deng et al. (2009)) is one such dataset containing 1000 classes over 14 million images. Each image requires individual labeling of all objects that it contains. It is possible to overcome this issue by gathering and curating data, though this is an expensive and time consuming task, especially when one considers the extra-labeling needed (i.e. drawing bounding boxes around objects) for object detection (Redmon et al. (2015), Russakovsky et al. (2014)). Proper datasets should contain millions of images, and each object within an image requires a box with an accurate label, either hand drawn or human verified (Russakovsky et al. (2014)). This can take hundreds of hours to complete (Deng et al. (2009)).

Models that require large amounts of data can be challenging in instances where there are few, or no, examples of a target class. The concept of open set recognition was introduced by Geng et al. (2021), in which the concept of "known-known classes" and "known-unknown" classes are introduced. Known-known classes are classes with labeled training data available, whereas known-unknown classes are those for which no positive training data are available. We will leverage this terminology throughout the paper.

One method of overcoming the difficulty of locating known-unknown classes is through the use of transfer learning. Transfer learning works by leveraging the information gained from previously trained models with known-known classes to an application with previously known-unknown classes, allowing the knowledge from the pre-trained model to bridge this gap (Konidaris & Barto (2006)). Another method for overcoming a lack of training data is low-shot learning. Low-shot methods attempt to address this by creating a model that needs as little as a single exposure to a new class (Radford et al. (2021); Wang et al. (2020)). Despite the utility of these methods in practice, these methods still require access to some instances of training examples.

Another approach to solving the issue of data availability is known as Zero-Shot Object Detection. This approach was first introduced by Bansal et al. (2018) and expanded upon by Zhu et al. (2020). These approaches attempt to use semantic similarity and multi-modal learning to detect previously unknown classes. Our approach differs from these in that they require end-to-end training for both bounding box predictions as well as classification, whereas our model does not require training the classifier. These methods also differ in that they require the use of specialized object detection datasets which include items that are semantically similar to detect previously unknown classes. This means that the datasets used to train these models must contain classes that closely resemble any unknown classes that they wish to detect. For example, in order to predict an object like a zebra, these models must be trained on objects which are similar, such as horses, giraffes, or cattle. Due to these requirements and because *DUCE* does not require semantic similarity during training, our model is considerably more robust.

Recently, CLIP (Radford et al. (2021)) introduced a zero-shot technique for image classification with performance competitive with task-specific supervised models. It was this development that inspired our work.

## 3 Methodology

*DUCE* combines the region proposal network and region of interest pooling methods used by FRCNN with the joint text/image embedding from CLIP. By doing so, it is able to realize zero-shot object detection capabilities. This design allows users to describe objects that they wish to find in images using natural language descriptions. See Figure 1 for the model architecture. Python code has been made publicly available. [1]

The model is built using the pre-trained image feature extractor from CLIP, which allows us to retain the ability to project images into the joint, CLIP feature representation space. The feature extractor utilizes the

---

[1] https://anonymous.4open.science/r/TORCH_CLIP_FRCNN_Trainable-2F46

ResNet-50 architecture (Radford et al. (2021)) and extracts features from images of any size and aspect ratio. This feature space is passed through a region proposal network (RPN) as described in Faster-RCNN (Ren et al. (2015)) to identify regions of interest (ROI's). Region Proposal Network parameters were unchanged from the paper defaults and produced anchor boxes of sizes 128, 256, 512 pixels, and aspect ratios of 0.5, 1.0, 2.0. The RPN then proposes up to 2000 regions of interest $\{P_1, P_2, ...P_n\}$ comprised of an objectness score and bounding box. ROI Pooling (Girshick (2015)), a method which uses max pooling to convert the features inside a region of interest into a small feature map with a fixed spatial extent, is performed on each proposal in order to transform them into the 7 x 7 x 1024 feature space before passing each transformed proposal through the pre-trained attention layer from the CLIP model. When complete, each proposal has been placed into the 1024 embedding dimension required by the CLIP model and can therefore be compared to the CLIP embedded text.

Simultaneously, a set of user provided, natural language textual inputs which describe objects of interest are each encoded into the joint space as 1024 length vectors $\{T_1, T_2, ...T_x\}$ using the pre-trained transformer model from CLIP Radford et al. (2021).

With the proposals and texts now embedded into the same joint space, we calculate the cosine similarity between each proposal and text, which is then normalized into a probability distribution via a softmax.

$$Q_n = S_c(P_n, T_x) \forall x \tag{1}$$

$$\mathrm{p}red = softmax(Q_n) \forall n \tag{2}$$

The text associated with the highest probability is the one that has been found to be most closely related to the proposal and so is the classification of that proposal. These classifications are joined with the previously computed bounding boxes from the RPN and reported as detections. Once the regions of interest are classified, the bounding boxes are clustered and consolidated before being presented to the user.

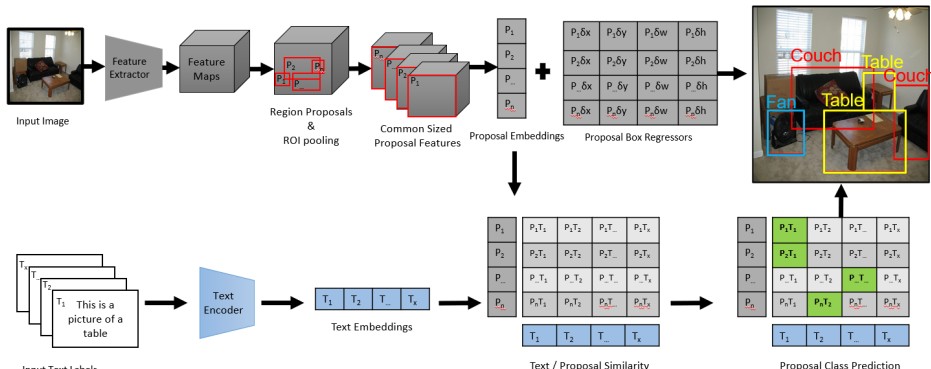

Figure 1: Model Architecture.

## 3.1 Box Clustering and Consolidation

Unlike a typical object detection model, whose classification confidence increases as the bounding box encompasses more of the object, this method is able to accurately predict class with high confidence given a region of interest that only captures a small section of the object. For example, the model is able to predict that an object is an airplane with high probability when provided with a region of interest that only includes the aircraft's engine. As a result, a conventional application of non-maximum suppression fails to consolidate and remove bounding boxes effectively. To correct for this, we introduce a novel method of *Box Clustering and Consolidation* (BCC). This approach utilizes DBSCAN (Ester et al. (1996)), a density based non-parametric clustering algorithm, to create per-class clusters from the midpoints from region predictions. Once the clusters are determined, we calculate the average of the bounding location and size weighted

by the class prediction confidence and return a consolidated list of predicted bounding box locations and classifications as shown in Algorithm 1.

---

**Algorithm 1** Box Clustering and Consolidation

---

bboxes ← [class_pred, class_prob, x_mid, y_mid, w, h]
**for** $unique(class\_pred) \in bboxes$ **do**
    db = DBSCAN($\epsilon$).fit(x_mid, y_mid)
    bboxes[cluster] = db.lables_
    **for** $cluster \in bboxes$ **do**
        $\overline{x_{mid}} = \mathbf{sum}$(x_mid × class_prob) / $\mathbf{sum}$ (class_prob)
        $\overline{y_{mid}} = \mathbf{sum}$(y_mid × class_prob) / $\mathbf{sum}$ (class_prob)
        $\overline{w} = \mathbf{sum}$(w × class_prob) / $\mathbf{sum}$ (class_prob)
        $\overline{h} = \mathbf{sum}$(h × class_prob) / $\mathbf{sum}$ (class_prob)
        class_conf = $\mathbf{max}$(class_prob)
        box_conf = $\mathbf{mean}$(class_prob)
        combined_boxes.append([class_pred, class_conf, box_conf, $\overline{x_{mid}}$, $\overline{y_{mid}}$, $\overline{w}$, $\overline{h}$])
    **end for**
**end for**
**Return** combined_boxes

---

BCC takes all instances of each unique predicted class and generates clusters from the box midpoints using DBSCAN. Once the predictions are clustered, we combine the predictions from each cluster into a single bounding box by taking the probability weighted average for the bounding box midpoint, width, and height. We also take the average of the class probabilities for each cluster as the bounding box confidence, and the maximum class probability as the classification confidence for the consolidated bounding box.

After performing BCC, Non-Maximum Suppression may be used to further suppress overlapping bounding boxes if required. One limitation of BCC is that we lose the ability to identify individual instances of closely grouped items, such as a school of fish. In these instances, BCC will provide one large bounding box containing all of the smaller regions.

### 3.1.1 Model Considerations

The intuition behind the model is simple. The CLIP architecture works for zero-shot image classification by predicting the relationship between an image and text (Radford et al. (2021)). It follows that CLIP would work equally well on cropped images. Because cropping and feeding segments of an image into CLIP directly would be slow and computationally expensive, another method is needed to reduce the time needed to perform this task, while still maintaining CLIP's pre-trained abilities. FRCNNs perform this task by producing one feature map for the entire image, then shares this convolutional feature map with the classifier through the use of use ROI pooling. However, a sliding window approach, would still be computationally expensive. By utilizing the RPN, we can reduce the number of regions for classification and maintain accuracy and zero-shot capability.

### 3.1.2 Best Practices

Because the model generates object detection probabilities based on the similarity between the object and user provided prompts, the number and quality of the labels is extremely important. If the user inputs a single text description, the model will report high probabilities for this class on most of the region proposals, even if they are only moderately related. As such, an increase in text descriptions results in increased model performance.

# 4 Results

In this section, we explore the performance of the model in comparison with state of the art object detection models by considering recall performance against known and unknown classes. We also explore the the sensitivity of BCC to its hyperparameters and its overall effect on the model via an ablation study.

## 4.1 Training

*DUCE* was trained on the COCO-2017 training dataset. The model only requires training the object and regression heads from the region proposal network, all other weights and biases are frozen. It is important to note that the model is never trained to classify objects, and instead is trained only to propose regions on which CLIP classification can be performed.

## 4.2 Hyper-Parameter Selection

*DUCE* allows for the adjustment of several parameters during region proposal and BCC. RPN hyperparameters include the number and shape of anchor boxes and RPN confidence. BCC introduces $\epsilon$ from DBSCAN Ester et al. (1996), and non-maximum suppression includes intersection-over-union. For the purposes of this paper, we maintained all hyper-parameters as the default values defined in their respective research and focused on the affects of $\epsilon$, which is the only new hyperparameter introduced by our method.

## 4.3 Inference

At test time, the model is provided with an image or images and a list of text descriptions of the classes the user intends to identify. These class descriptions can be as short as a single word or as long as several sentences.

## 4.4 Experiments

*DUCE* was evaluated on a selection of fourteen known classes shared with the the COCO dataset on which all of the models were trained and five unknown classes. The classes listed in Appendix A were selected to minimize potential class bleed thus ensuring that the zero-shot evaluations were accurate. In instances where there were multiple ImageNet labels for the same COCO class, all ImageNet labels were combined into the single COCO label. For balance, the dataset was subsampled to ensure equal representation of all classes. We tested *DUCE* with and without BCC. All models tested applied greedy non-maximum suppression with an intersection over union threshold of 0.5 unless otherwise noted. We follow Bansal et al. (2018) and Lu et al. (2016) and evaluate the models based on recall. *DUCE* was provided with the phrase "this is a photo of a __" for each COCO class label in the dataset.

## 4.5 Generalized Zero-shot Performance

Tables 1 and 2 shows a comparison of *DUCE*, YOLOv5, and FRCNN results for two IoU thresholds, for both known and unknown classes. Overall, results for the known-known classes were generally comparable across models, but the FRCNN and YOLO models cannot classify any instances of the known-unknown classes. Comparatively, our model has reliable results regarding the known-unknown classes.

For an IoU of 0.5, FRCNN achieves a 0.8 recall on the known classes. For the same parameters, YOLOv5 produces a 0.56 recall on known classes. Neither model was capable of classifying unknown classes. *DUCE* without BCC, achieves a recall of 0.56 on the known classes and 0.48 on the unknown classes. This performance is behind the FRCNN, but matches the performance of YOLOv5 on the known classes. However, *DUCE* is able to achieve similar results on unknown classes as well.

When performance is assessed at an IoU of 0.1, the performance of *DUCE* increases dramatically, compared to the increased performance by the FRCNN and YOLOv5. This is because *DUCE* is able to produce high confidence detections on both small and large region proposals relative to the ground truth bounding box, for which the IoU is small.

Table 1: Recall Results at IOU = .1

| | Class | IOU @ .1 | | | |
| --- | --- | --- | --- | --- | --- |
| | | *FRCNN* | *Yolov5* | *DUCE* | *DUCE With BCC* |
| | clock | **0.84** | 0.33 | 0.52 | 0.51 |
| | vase | **0.9** | 0.67 | 0.54 | 0.5 |
| | toaster | **0.53** | 0.02 | 0.36 | 0.25 |
| | microwave | **0.79** | 0.53 | 0.61 | 0.51 |
| | mouse | **0.8** | 0.57 | 0.62 | 0.35 |
| | potted plant | **0.81** | 0.27 | 0.78 | 0.41 |
| Known Knowns | sports ball | **0.68** | 0.44 | 0.67 | 0.45 |
| | zebra | **1** | 0.94 | 0.99 | 0.76 |
| | dog | **0.99** | 0.78 | **0.99** | 0.86 |
| | bird | **0.97** | 0.77 | 0.93 | 0.88 |
| | bench | **0.96** | 0.72 | 0.84 | 0.6 |
| | parking meter | 0.93 | 0.74 | **0.95** | 0.71 |
| | airplane | **1** | 0.93 | 0.99 | 0.93 |
| | bicycle | **0.95** | 0.8 | **0.95** | 0.78 |
| | | | | | |
| | Average | **.87** | .61 | .77 | .61 |
| | | | | | |
| | lizard | 0.00 | 0.00 | **0.62** | 0.56 |
| | turtle | 0.00 | 0.00 | **0.55** | 0.52 |
| Known Unknowns | pen | 0.00 | 0.00 | **0.95** | 0.71 |
| | cowboy hat | 0.00 | 0.00 | **0.76** | 0.46 |
| | tank | 0.00 | 0.00 | **0.99** | 0.87 |
| | | | | | |
| | Average | 0.00 | 0.00 | **.77** | .62 |

Recall scores are separated by class type, known-known vs known-unknown. Average scores for each class type are included.

An interesting note is that *DUCE* achieves 0.08 worse recall on unknown classes at IoU of 0.5 but is equal at IoU of 0.1. This can be explained by recognizing that the region proposal network is the only portion of the model that is trained. Although the RPN is trained on certain classes, it still predicts objects which are outside of its training set, however these predictions contain poor quality bounding boxes. for objects within its training dataset, the bounding boxes are of higher quality, resulting in better performance at higher IoU.

*DUCE* also compares favorably to the models proposed by Bansal et al. (2018) and Zhu et al. (2020). Bansal et al. (2018) reports a best recall of 0.27 for known and unknown classes at IoU 0.5. Comparatively, *DUCE* achieves a recall of 0.25 with BCC and 0.54 without BCC. Zhu et al. (2020) use mAP as their preferred metric and their model is able to achieve .04 for known-unknown classes at an IoU of 0.5. *DUCE* achieves a mAP of 0.08 for known-unknown classes at an IoU of 0.5.

### 4.5.1 Hyperparameter Sensitivity

In this paper, we introduce BCC, a method of bounding box consolidation for object detection where many high confidence bounding boxes exist. As a part of this study, we explore the space over which this hyperparameter is valid. $\epsilon$ (Eps-neighborhood) is a method by which clusters can be created by defining the maximum distance between points able to be considered within the same cluster (Ester et al. (1996)). We consider each image an m x n pixel Cartesian space in which bounding boxes are defined by their midpoints. These midpoints are used in the DBSCAN algorithm to find clusters. As such, the sizes of each object within the dataset will affect the most appropriate value for this hyperparameter. Because the Imagenet dataset

Table 2: Recall Results at IOU = .5

| | | IOU @ .5 | | | |
|---|---|---|---|---|---|
| | Class | *FRCNN* | *Yolov5* | *DUCE* | *DUCE With BCC* |
| | clock | **0.68** | 0.27 | 0.43 | 0.24 |
| | vase | **0.9** | 0.64 | 0.37 | 0.23 |
| | toaster | **0.52** | 0.02 | 0.11 | 0.06 |
| | microwave | **0.75** | 0.51 | 0.3 | 0.14 |
| | mouse | **0.75** | 0.54 | 0.25 | 0.11 |
| | potted plant | **0.48** | 0.16 | 0.37 | 0.12 |
| Known Knowns | sports ball | **0.66** | 0.41 | 0.36 | 0.16 |
| | zebra | **0.97** | 0.9 | 0.9 | 0.31 |
| | dog | **0.97** | 0.77 | 0.89 | 0.64 |
| | bird | **0.96** | 0.72 | 0.78 | 0.52 |
| | bench | **0.91** | 0.69 | 0.65 | 0.21 |
| | parking meter | **0.79** | 0.55 | 0.62 | 0.25 |
| | airplane | **0.96** | 0.91 | **0.96** | 0.23 |
| | bicycle | **0.91** | 0.72 | 0.81 | 0.3 |
| | | | | | |
| | Average | **.8** | .56 | .56 | .25 |
| | | | | | |
| | lizard | 0.00 | 0.00 | **0.38** | 0.13 |
| | turtle | 0.00 | 0.00 | **0.36** | 0.27 |
| Known Unknowns | pen | 0.00 | 0.00 | **0.56** | 0.22 |
| | cowboy hat | 0.00 | 0.00 | **0.28** | 0.11 |
| | tank | 0.00 | 0.00 | **0.83** | 0.49 |
| | | | | | |
| | Average | 0.00 | 0.00 | **.48** | .24 |

Recall scores are separated by class type, known-known vs known-unknown. Average scores for each class type are included.

contains objects of many varying sizes, mAP and recall were calculated for a range of $\epsilon$ between 5 and 50. We find that the model is robust to $\epsilon$ in the range of values from 5 to 50 as shown in Figure 2.

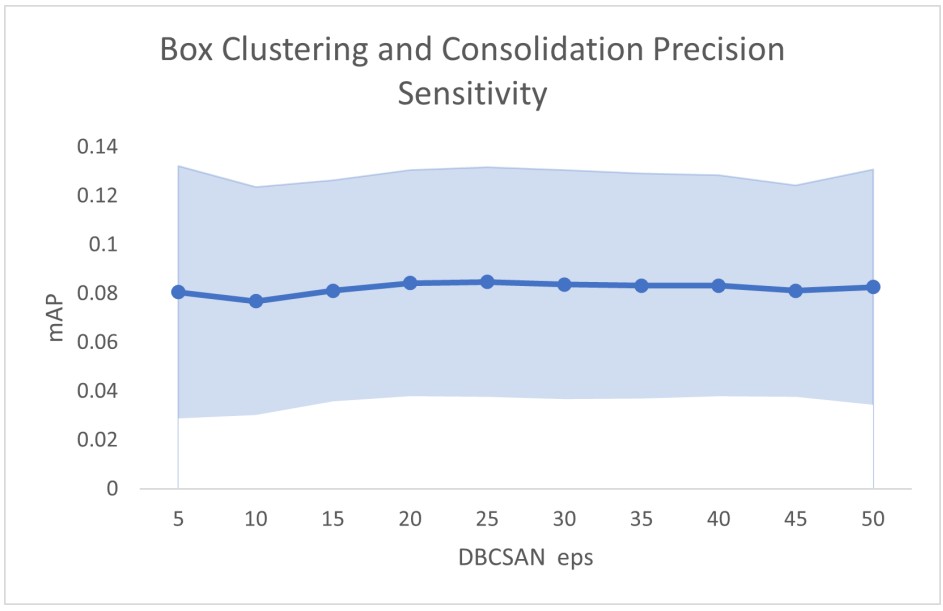

Figure 2: Box Consolidation and Clustering Sensitivity. (Shaded region indicates standard deviation)

### 4.5.2  Ablation Study

In Tables 1 through 4 we compare performance with BCC ($\epsilon = 35$) and without BCC across IoU values of 0.1 and 0.5. These results show moderate increases in mAP at the expense of recall. The loss of recall when using this method can be attributed to the clustering method consolidating the bounding boxes from many small images into one large one. An example of this is demonstrated in Figure 4. The results of the method are dramatic, however as shown in Figure 3.

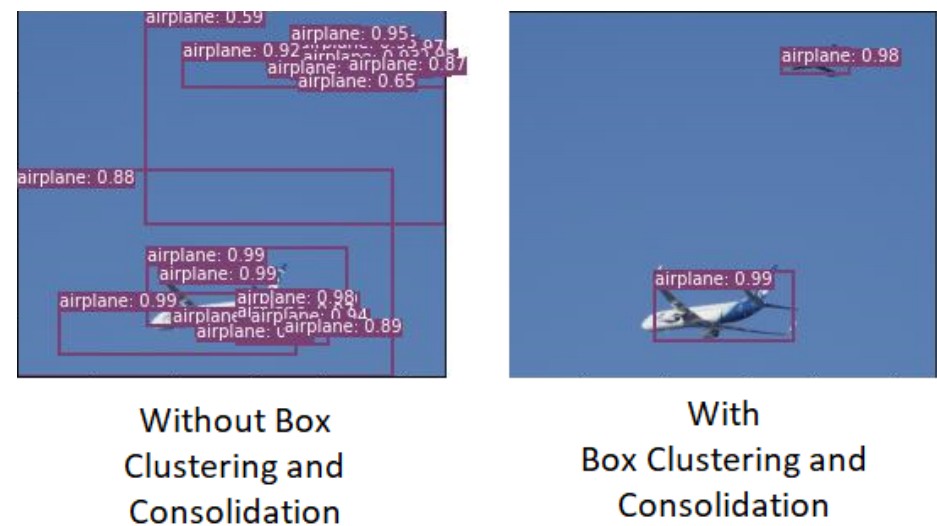

Figure 3: Box Consolidation and Clustering Ablation.

Table 3: Precision Results at IOU = .1

| | Class | IOU @ .1 | | | |
|---|---|---|---|---|---|
| | | **FRCNN** | **Yolov5** | **DUCE** | **DUCE With BCC** |
| | clock | 0.24 | **0.88** | 0.02 | 0.32 |
| | vase | 0.17 | **0.6** | 0.04 | 0.38 |
| | toaster | 0.52 | **1** | 0.03 | 0.14 |
| | microwave | 0.31 | **0.86** | 0.02 | 0.23 |
| | mouse | 0.36 | **0.79** | 0.01 | 0.04 |
| | potted plant | 0.16 | **0.44** | 0.03 | 0.21 |
| Known Knowns | sports ball | 0.5 | **0.93** | 0.02 | 0.14 |
| | zebra | 0.32 | **0.78** | 0.03 | 0.28 |
| | dog | 0.38 | **0.92** | 0.02 | 0.15 |
| | bird | 0.23 | **0.75** | 0.02 | 0.18 |
| | bench | 0.15 | **0.82** | 0.01 | 0.15 |
| | parking meter | 0.37 | **0.81** | 0.02 | 0.19 |
| | airplane | 0.32 | **0.93** | 0.01 | 0.22 |
| | bicycle | 0.14 | **0.8** | 0.02 | 0.32 |
| | | | | | |
| | Average | .3 | **.81** | .02 | .21 |
| | | | | | |
| | lizard | 0.00 | 0.00 | 0.07 | **0.41** |
| | turtle | 0.00 | 0.00 | 0.06 | **0.31** |
| Known Unknowns | pen | 0.00 | 0.00 | 0.04 | **0.16** |
| | cowboy hat | 0.00 | 0.00 | 0.06 | **0.19** |
| | tank | 0.00 | 0.00 | 0.02 | **0.1** |
| | | | | | |
| | Average | 0.00 | 0.00 | .05 | **.23** |

The precision results for each model separated by class type and known-known vs known-unknown. Average scores for each type are included.

Table 4: Precision Results at IOU = .5

| | | IOU @ .5 | | | |
|---|---|---|---|---|---|
| | Class | *FRCNN* | *Yolov5* | *DUCE* | *DUCE With BCC* |
| | clock | 0.2 | **0.71** | 0.02 | 0.15 |
| | vase | 0.17 | **0.58** | 0.03 | 0.18 |
| | toaster | 0.51 | **1** | 0.01 | 0.03 |
| | microwave | 0.29 | **0.83** | 0.01 | 0.06 |
| | mouse | 0.34 | **0.74** | 0 | 0.01 |
| | potted plant | 0.09 | **0.25** | 0.02 | 0.06 |
| Known Knowns | sports ball | 0.49 | **0.89** | 0.01 | 0.05 |
| | zebra | 0.31 | **0.75** | 0.03 | 0.11 |
| | dog | 0.37 | **0.9** | 0.02 | 0.11 |
| | bird | 0.22 | **0.7** | 0.02 | 0.11 |
| | bench | 0.14 | **0.77** | 0.01 | 0.05 |
| | parking meter | 0.31 | **0.61** | 0.01 | 0.07 |
| | airplane | 0.31 | **0.91** | 0.01 | 0.06 |
| | bicycle | 0.14 | **0.72** | 0.02 | 0.12 |
| | | | | | |
| | Average | .28 | **.74** | .02 | .08 |
| | | | | | |
| | lizard | 0.00 | 0.00 | 0.04 | **0.09** |
| | turtle | 0.00 | 0.00 | 0.04 | **0.16** |
| Known Unknowns | pen | 0.00 | 0.00 | 0.02 | **0.05** |
| | cowboy hat | 0.00 | 0.00 | 0.02 | **0.05** |
| | tank | 0.00 | 0.00 | 0.01 | **0.06** |
| | | | | | |
| | Average | 0.00 | 0.00 | .03 | **.08** |

The precision results for each model separated by class type and known-known vs known-unknown. Average scores for each type are included.

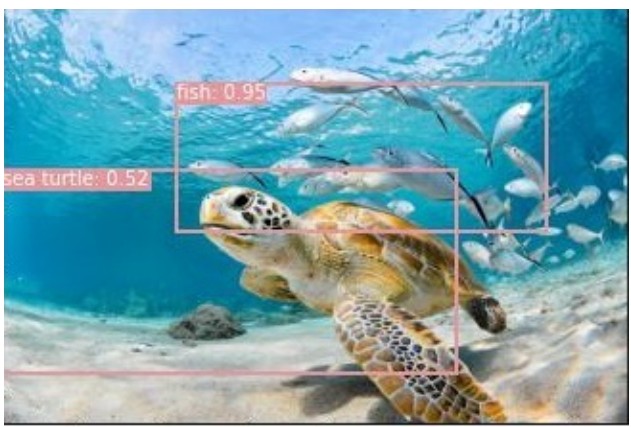

Figure 4: Box Consolidation and Clustering Limitations.

### 4.6 Conclusion

In this paper, we introduce *DUCE*, a novel method of zero-shot object detection requiring little training through the combination of an RPN and the multi-modal embeddings from CLIP, and BCC, a bounding box suppression method utilizing the unsupervised clustering technique DBSCAN. *DUCE* shows promising results, matching the recall performance from YOLOv5 and being within 10 percent of FRCNN on known classes and now state of the art 77 percent recall on unknown classes. This paper also explores the effects of BCC and its hyperparameter $\epsilon$ and finds that, while the optimal selection is based on object size within an image, it is generally stable within the range of 5 - 50. We also explore how the addition of BCC affects model precision, by increasing the average precision from approximately 5% to 23%.

### 4.7 Limitations

Current state of the art object detectors perform bounding box regression based on the object class, that is, each class learns its own set of regressors. This architecture allows these models to learn nuances about the size and aspect ratio for each class in order to draw tight bounding boxes around the objects. Since *DUCE* must predict bounding boxes for objects on which it has never been trained, it cannot use bounding box regressors in this way and is therefore unable to match the performance of state of the art models on known-known classes in this regard. However, *DUCE* achieves consistent performance across known-known and known-unknown classes, while the state of the art model cannot predict any known-unknown classes. Additionally, the ability of of the CLIP classifier to accurately classify objects given only a small portion of the object with high accuracy presents challenges in evaluating the model using standard object detection measures such as mean Average Precision as shown in Figure 5.

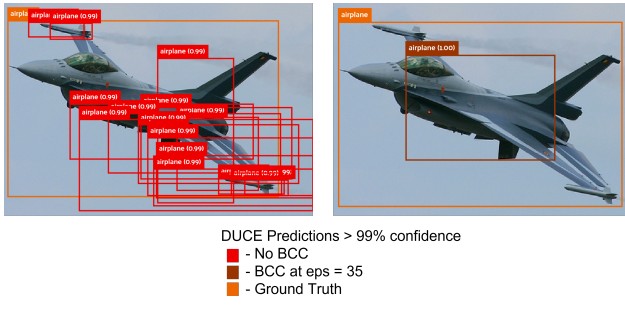

Figure 5: Evaluation Difficulty

Using standard evaluation metrics, all of the proposed bounding boxes from this image are considered false positive

Furthermore, because *DUCE* compares the similarity of text and regions from the proposed region, performance is impacted by the text prompts. This issue is also discussed by Radford et al. (2021).

Lastly, BCC is sensitive to $\epsilon$, and when selected must consider the relative size of the objects within the image. When a large $\epsilon$ is selected, it will cluster any instances of small objects which appear in groups.

### 4.8 Future Work

As an overarching theme, we believe future work should be aimed at improving the model's ability to serve as a zero-shot object detector. There are several possible avenues to do this, given the results of this study.

First, add more information in the textual description, specifically related to the context of the objects in the images when training the CLIP classification model from which *DUCE* is derived. Information should focus on describing both the background and profile of the image. For instance, "this is a picture is a side profile of a tank in the woods." This helps to give the model more context when classifying the image regardless of the environment. This method may require the collection of more training data to fine-tune a CLIP model with more information on the profile and background.

Another method for improvement would be to utilize a pre-trained language model as the CLIP text embedding method. CLIP uses a basic transformer and has limited capability to encode words that it has not been previously exposed to. A model that can understand language, context, and a wider variety of words would be better suited to handle new and previously unseen text, and would therefore improve detection performance.

Lastly, another possible avenue of future work is to build a "no-train" model. This could be done by removing the RPN entirely and replace it with a sliding window approach proposed by Dalal & Triggs (2005). By using BCC, we can forgo the need to create good bounding boxes, rather a cluster of boxes can be manipulated to draw a "good enough" box around the object. While this may not yield the highest scores in a testing environment, when faced with the challenges of real world data, this approach would allow a user to specify an object to search for in real time, and the model would be able to point the user to an area of interest within an image or video feed.

### Broader Impact Statement

Many of the ethical considerations of this model are shared by Radford et al. (2021) with regard to their CLIP model. There are positive unexpected outcomes of the CLIP embedding model. For instance, improvements in accuracy for racial, gender, and age classification were seen in CLIP models when compared to other classification models that focused solely on these categories Radford et al. (2021). But there are also negative impacts. Radford et al. (2021) state that "CLIP is trained on text paired with images on the internet. These image-text pairs are unfiltered and uncurated and result in CLIP models learning many social biases." Because the classification portion of our model relies heavily on the training and data that the CLIP classification was exposed to, it is likely that CLIP introduces biases in the relationships between objects in our model. This raises potential ethical considerations if the model associates objects in an inaccurate or stereotypical fashion. An example of this transfer of social bias can be seen when our model is given a picture of sports equipment but only given labels according to race such as "black man" or "white man". The model will classify the equipment and according to how closely the image embeds to either label, with clear racial stereotypes regarding sports (i.e. boxing gloves receive the label of "Black Man").

Additionally, *DUCE* has shown the capability to detect people in images by name as demonstrated in Figure 6. Because the CLIP model was trained on a dataset of 400 million (image, text) pairs collected from the internet, the by-name labeling will be most accurate for individuals with multiple instances for CLIP to train on. Based on the nature of social media, the most frequently identified individuals on Instagram are celebrities. In Figure 6, the model was first given a test photo with only the labels for genders man and woman. The results were that the model labeled the male celebrity as man with a confidence of .58 and the female celebrity as woman with a confidence of .53. Then, the model was given the same test photo with labels for their proper names: Bradley Cooper and Jennifer Lawrence. The result was that Bradley Cooper was identified and labeled correctly with a confidence of .98 and Jennifer Lawrence was identified

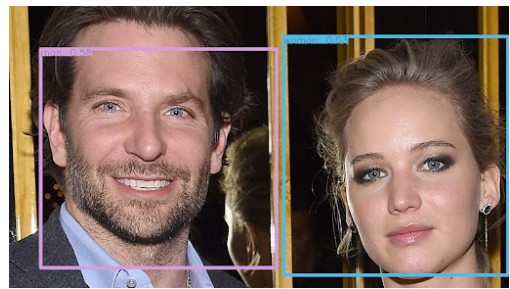
(a) Output with Gender Labels

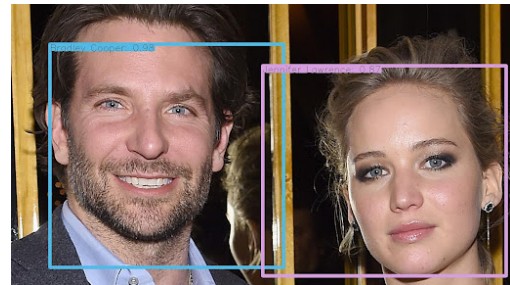
(b) Output with Celebrity Name Labels

Figure 6: The model was given a photo of two celebrities. In 6a, the only labels provided were genders ("man", "woman"). In 6b, the only labels provided were the correct names for each celebrity.

and labeled correctly with a confidence of .87. This raises privacy concerns in addition to questions about how models like these work to classify items.

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

## A  Class Selection

Below are tables showing variables selected to evaluate model performance. These variables included fourteen known-known classes, and five known-unknown classes(Geng et al. (2021)). ImageNet classes were combined into super classes and labeled to match MS COCO classes. The models in this paper were trained on the MS COCO dataset, but evaluated on corresponding and unknown classes from the ImageNet dataset, this allowed for zero-shot evaluation on an established dataset.

Table 5: Selected Known-Known Classes

| MS COCO Label | ImageNet Label |
|---|---|
| bicycle | 'bicycle-built-for-two, tandem bicycle, tandem ', 'mountain bike, all-terrain bike, off-roader ' |
| airplane | 'airliner' |
| parking meter | 'parking meter' |
| bench | 'park bench' |

**Table 5 continued from previous page**

| MS COCO Label | ImageNet Label |
|---|---|
| bird | 'bulbul ', 'jay ', 'magpie ', 'chickadee ', 'water ouzel, dipper ', 'goldfinch, Carduelis carduelis ', 'house finch, linnet, Carpodacus mexicanus ', 'junco, snowbird ', 'indigo bunting, indigo finch, indigo bird, Passerina cyanea ', 'robin, American robin, Turdus migratorius ', 'black grouse ', 'ptarmigan ', 'ruffed grouse, partridge, Bonasa umbellus ', 'prairie chicken, prairie grouse, prairie fowl ', 'peacock ', 'quail ', 'partridge ', 'African grey, African gray, Psittacus erithacus ', 'macaw ', 'sulphur-crested cockatoo, Kakatoe galerita, Cacatua galerita ', 'lorikeet ', 'coucal ', 'bee eater ', 'hornbill ', 'hummingbird ', 'jacamar ', 'toucan ', 'drake ', 'red-breasted merganser, Mergus serrator ', 'goose ', 'black swan, Cygnus atratus ', 'white stork, Ciconia ciconia ', 'black stork, Ciconia nigra ', 'spoonbill ', 'flamingo ', 'little blue heron, Egretta caerulea ', 'American egret, great white heron, Egretta albus ', 'bittern ', 'crane ', 'limpkin, Aramus pictus ', 'European gallinule, Porphyrio porphyrio ', 'American coot, marsh hen, mud hen, water hen, Fulica americana ', 'bustard ', 'ruddy turnstone, Arenaria interpres ', 'kite ', 'great grey owl, great gray owl, Strix nebulosa ', 'bald eagle, American eagle, Haliaeetus leucocephalus ', 'brambling, Fringilla montifringilla ', 'vulture ', 'ostrich, Struthio camelus ', 'hen ', 'cock ', 'king penguin, Aptenodytes patagonica ', 'pelican ', 'dowitcher ', 'oystercatcher, oyster catcher ', 'albatross, mollymawk ' |

**Table 5 continued from previous page**

| MS COCO Label | ImageNet Label |
|---|---|
| dog | 'Maltese dog, Maltese terrier, Maltese ', 'Chihuahua ', 'Japanese spaniel ', 'Pekinese, Pekingese, Peke ', 'Shih-Tzu ', 'Blenheim spaniel ', 'papillon ', 'toy terrier ', 'Rhodesian ridgeback ', 'Sealyham terrier, Sealyham ', 'beagle ', 'Afghan hound, Afghan ', 'basset, basset hound ', 'bloodhound, sleuthhound ', 'bluetick ', 'black-and-tan coonhound ', 'Walker hound, Walker foxhound ', 'English foxhound ', 'redbone ', 'Lakeland terrier ', 'Irish wolfhound ', 'borzoi, Russian wolfhound ', 'Italian greyhound ', 'whippet ', 'Ibizan hound, Ibizan Podenco ', 'Norwegian elkhound, elkhound ', 'otterhound, otter hound ', 'Saluki, gazelle hound ', 'Scottish deerhound, deerhound ', 'wire-haired fox terrier ', 'Weimaraner ', 'Staffordshire bullterrier, Staffordshire bull terrier ', 'American Staffordshire terrier, Staffordshire terrier, American pit bull terrier, pit bull terrier ', 'Bedlington terrier ', 'Border terrier ', 'Kerry blue terrier ', 'Irish terrier ', 'Norfolk terrier ', 'Norwich terrier ', 'Yorkshire terrier ', 'Airedale, Airedale terrier ', 'cairn, cairn terrier ', 'Australian terrier ', 'Dandie Dinmont, Dandie Dinmont terrier ', 'Boston bull, Boston terrier ', 'miniature schnauzer ', 'giant schnauzer ', 'standard schnauzer ', 'Scotch terrier, Scottish terrier, Scottie ', 'Tibetan terrier, chrysanthemum dog ', 'soft-coated wheaten terrier ', 'silky terrier, Sydney silky ', 'West Highland white terrier ', 'Lhasa, Lhasa apso ', 'flat-coated retriever ', 'curly-coated retriever ', 'golden retriever ', 'Labrador retriever ', 'Chesapeake Bay retriever ', 'German short-haired pointer ', 'Rottweiler ', 'German shepherd, German shepherd dog, German police dog, alsatian ', 'Doberman, Doberman pinscher ', 'miniature pinscher ', 'English setter ', 'vizsla, Hungarian pointer ', 'Irish setter, red setter ', 'Gordon setter ', 'Brittany spaniel ', 'clumber, clumber spaniel ', 'English springer, English springer spaniel ', 'Welsh springer spaniel ', 'cocker spaniel, English cocker spaniel, cocker ', 'Sussex spaniel ', 'kuvasz ', 'Irish water spaniel ', 'schipperke ', 'groenendael ', 'malinois ', 'briard ', 'kelpie ', 'komondor ', 'Old English sheepdog, bobtail ', 'Shetland sheepdog, Shetland sheep dog, Shetland ', 'timber wolf, grey wolf, gray wolf, Canis lupus ', 'Pembroke, Pembroke Welsh corgi ', 'Cardigan, Cardigan Welsh corgi ', 'toy poodle ', 'miniature poodle ', 'standard poodle ', 'Mexican hairless ', 'affenpinscher, monkey pinscher, monkey dog ', 'basenji ', 'Leonberg ', 'Newfoundland, Newfoundland dog ', 'Great Dane ', 'Saint Bernard, St Bernard ', 'French bulldog ', 'Tibetan mastiff ', 'bull mastiff ', 'boxer ', 'EntleBucher ', 'Appenzeller ', 'Greater Swiss Mountain dog ', 'collie ', 'Eskimo dog, husky ', 'malamute, malemute, Alaskan malamute ', 'Siberian husky ', 'dalmatian, coach dog, carriage dog ', 'pug, pug-dog ', 'Great Pyrenees ', 'Samoyed, Samoyede ', 'Pomeranian ', 'Bernese mountain dog ', 'Border collie ', 'African hunting dog, hyena dog, Cape hunting dog, Lycaon pictus ', 'chow, chow chow ', 'dhole, Cuon alpinus ', 'keeshond ', 'Brabancon griffon ', 'dingo, warrigal, warragal, Canis dingo ', 'coyote, prairie wolf, brush wolf, Canis latrans ', 'red wolf, maned wolf, Canis rufus, Canis niger ', 'white wolf, Arctic wolf, Canis lupus tundrarum ', 'Bouvier des Flandres, Bouviers des Flandres ' |
| zebra | 'zebra' |
| sports ball | 'baseball ', 'basketball ', 'croquet ball ', 'golf ball ', 'ping-pong ball ', 'rugby ball ', 'soccer ball ', 'tennis ball ', 'volleyball ' |
| potted plant | 'pot, flowerpot ' |
| mouse | 'mouse, computer mouse ' |
| microwave | 'microwave, microwave oven ' |
| toaster | 'toaster ' |
| clock | 'analog clock ', 'digital clock ', 'digital watch ', 'wall clock ' |
| vase | 'vase' |

Table 6: Selected Known-Unknown Classes

| MS COCO Label | ImageNet Label |
|---|---|
| lizard | 'banded gecko ', 'common iguana, iguana, Iguana iguana ', 'American chameleon, anole, Anolis carolinensis ', 'whiptail, whiptail lizard ', 'Komodo dragon, Komodo lizard, dragon lizard, giant lizard, Varanus komodoensis ', 'African chameleon, Chamaeleo chamaeleon ', 'green lizard, Lacerta viridis ', 'Gila monster, Heloderma suspectum ', 'alligator lizard ', 'frilled lizard, Chlamydosaurus kingi ', 'agama ', 'European fire salamander, Salamandra salamandra ', 'common newt, Triturus vulgaris ', 'eft ', 'spotted salamander, Ambystoma maculatum ', 'axolotl, mud puppy, Ambystoma mexicanum ' |
| turtle | 'loggerhead, loggerhead turtle, Caretta caretta ', 'leatherback turtle, leatherback, leathery turtle, Dermochelys coriacea ', 'mud turtle ', 'terrapin ', 'box turtle, box tortoise ' |
| pen | 'fountain pen ', 'ballpoint, ballpoint pen, ballpen, Biro ' |
| cowboy hat | 'cowboy hat, ten-gallon hat ' |
| tank | 'tank, army tank, armored combat vehicle, armoured combat vehicle ' |

