# OpenReview forum: "Zero-shot Object Detection with a Text and Image Contrastive Model"
_TMLR — Withdrawn by Authors_

### Review · Reviewer_vd63 · 2022-06-06

**Summary Of Contributions:**

The paper proposes a zero-shot object detection method, Detection of Unknown objects through CLIP Extraction (DUCE). DUCE builds upon the zero-shot classification capabilities of CLIP and the object-agnostic region proposals from Faster R-CNN. Essentially, DUCE applies ROI pooling based on the RPN on the final feature map of CLIP. The CLIP scores of each proposal against a given set of prompts including classes of interest return the set of detections in the image. Importantly, the pre-trained CLIP is never fine-tuned; it is only the RPN that is tuned for the dataset of interest.

DUCE is tested on a subset of COCO-2017. The evaluation uses 14 classes ("known") on which the RPN is trained and 5 classes ("unknown") that are never seen during training. DUCE is compared against fully-supervised methods like FRCNN and YOLOv5 in terms of the recall and precision values.

**Broader Impact Concerns:**

I agree with the authors' observation that their method will inherit the biases and privacy concerns in the CLIP models. The statement could have been stronger had they discussed ways to address such issues in the future - in the context of zero-shot object detection.

**Requested Changes:**

This echoes the points in the weaknesses.

* Please discuss the missing reference above and discuss how their work is significantly different from the ICLR 2022 paper.
* Please present fair empirical evidence that BCC really does improve the object detection performance.
* Please report the performances with more holistic detection evaluation metrics - mAP or PR curves.
* Please explain how and why the known and unknown classes are chosen.
* Please work on the sectioning and presentation throughout the paper.

**Strengths And Weaknesses:**

## Strengths

The authors have proposed a conceptually working solution for zero-shot object detection. The method smartly takes advantage of the zero-shot classification capabilities of the CLIP model. T


## Weaknesses

* Missing reference

I have come across the following work published and presented at ICLR 2022: https://openreview.net/pdf?id=lL3lnMbR4WU
This paper has also demonstrated zero-shot object detection based on CLIP. I believe the submission must discuss this paper and compare how their work is significantly different from this work.

* Is BCC a contribution?

BCC, Box Clustering and Consolidation method, is stressed throughout the paper as a "novel" component that is effective. I do not find any experimental backup for the effectiveness. Adding BCC on DUCE seems to decrease the recall (Table 1&2) while increasing the precision (Table 3&4). The discussion in Sec 4.5.2 is also somewhat ambivalent. What is the conclusion? Should we use the BCC or not?

It is also unfair to tune BCC hyperparameters when the baseline non-maximal suppression (NMS) threshold is fixed.

* Why show precision and recall separately?

It is sometimes possible to set the operating threshold such that the precision and recall values generate conclusions that serves a specific conclusion. That's why people report the precision-recall curves over the range of operating thresholds and report the mean average precision (mAP) values as the summary statistics. Why did the authors not follow this practice and instead split the precision and recall reports across Tables 1-4?

* How are the "known" and "unknown" classes chosen?

The 14 known classes and 5 unknown classes are a very specific choice of classes among the 80+ COCO classes. The experimental conclusion may change depending on the choice of classes to test. Can authors elaborate on the claim: "The classes [...] were selected to minimize potential class bleed". There is not only one set of classes that satisfies the statement above. Why this particular choice of classes?

* Sectioning is not ideal.

The method section (sec 3) starts with the description of the method. Then it explains the BCC algorithm in Sec 3.1. Then it comes back to the DUCE model in Sec 3.1.1 and 3.1.2. I do not get the logic behind the sectioning. Section 4 is likewise ineffectively organised.

* Presentation

The figures and tables need more work to be better readable and presentable in general.

---

### Review · Reviewer_Xxpb · 2022-06-07

**Summary Of Contributions:**

This paper considers the problem of generalizable zero-shot object detector. To solve this problem, the authors leverage the RPN and CLIP to construct the zero-shot detector. Moreover, a new bounding box suppression approach (BCC) is proposed to handle the specific situations caused by the proposed detector. Experiments on COCO-2017 show the benefit of the proposed compared to FRCNN and YOLO5.

**Broader Impact Concerns:**

The authors have well addressed the broader impact concerns.

**Requested Changes:**

1. Rewritten the overall paper: highlighting the benefit of the zero-shot detection and the motivation of the proposed.

2. Comparison with the closely related work [A].

3. Explain the benefit of BCC.

4. Explain why NMS suffers from the proposed method and why BCC can achieve better results.

5. More visualizations of results.

The details can be found in the weakness section.

**Strengths And Weaknesses:**

--------Strengths--------
+ This paper considers an interesting and challenging problem and proposes an effective zero-shot object detector is presented.

+ Source code is provided for reproducing.

+ Limitation of the proposed method is well discussed, helping readers better understand the proposed method.


--------Weaknesses--------

- My first major concern is the writing of this paper. This paper is not well-written and not well-structured. For example, this paper considers a new problem in object detection, however, in the introduction, there is no discussion about the benefit and the importance of the new setting.

- My second concern is the novelty of this paper. It should be noticed that the zero-shot detection using CLIP is already studied in the community [A]. It uses CLIP to perform zero-shot novel class detection and it also proposes effective approaches based on the CLIP-based baseline. However, the proposed method seems very similar to this simple baseline. That is, zero-shot detection is not new and using CLIP to address this problem is also not new.

- My third concern is missing important work. As pointed the above, [A] is proposed to use CLIP to solve the zero-shot detection. However, this paper does not provide any discussion about it. In addition, the comparison with it is missed in the experiments. Thus, without comparing it, I can not agree with the novelty and the advantage of the proposed method.

- This paper proposes the BCC. However, from Table 1-3, I found that BCC achieves poor results. Please explain this.

- Moreover, the drawback of NMS is not very clear to me under the proposed method. Please provide a more detailed explanation of why NMS meets the problem and why the proposed BCC can solve this issue.

- More visualizations of the detection results should be provided to verify the effectiveness of the proposed method.

[A] Open-vocabulary Object Detection via Vision and Language Knowledge Distillation. ICLR 2022.

---

### Review · Reviewer_ohGv · 2022-06-08

**Summary Of Contributions:**

This work argues two major contributions for zero-shot object detection. First, it introduces the DUCE detector leveraging the zero-shot performance of CLIP (Radford et al. (2021)). Second, it introduces BCC as a new approach to replace non-maximum suppression.


**Broader Impact Concerns:**

No specific ethical concern in this work.


**Requested Changes:**

Please see the comments in Weaknesses.
This paper contains too many flaws in various aspects.


**Strengths And Weaknesses:**

<Weaknesses>

1. This work fails to identify its main contribution.
- Section 1 (introduction) is too short and fails to mention what contributions this work delivers.
- In section 2, it ignores a large body of previous work on zero-shot object detection, to name a few:
- Zheng et al. Visual Language Based Succinct Zero-Shot Object Detection, ACMMM2021.
- Bansal et al. Zero-Shot Object Detection, ECCV 2020.
- Zheng et al. Zero-Shot Object Detection with Transformers, ICIP 2021.

2. No technical novelty is clarified.
- This work introduces two novel models named DUDE and BCC, which are simple adoption of well-known techniques.
- DUDE is largely based on CLIP with no meaningful update.
- BCC is a simple application of DBSCAN to the overlapped bounding boxes.

3. Experiment evaluations are not convincing.
- Only a single dataset (COCO-2017) is tested.
- No meaningful baseline is compared.
- The performance of DUDE and BCC is poor in normal object detection tasks.

---

### Note · Authors · 2022-06-14

**Comment:**

I would like to thank the reviewers for their time and comments on our paper.  We were unaware of the paper "OPEN-VOCABULARY OBJECT DETECTION VIA VISION AND LANGUAGE KNOWLEDGE DISTILLATION" presented at ICLR 2022, and after review do not believe that our findings are significantly different from theirs.  We are therefore withdrawing our paper from consideration

We will continue our work with these methods and hope to present new findings in our future work.

Once again, we are very grateful for the reviewers comments and time and will incorporate them into our future work as necessary.

**Withdrawal Confirmation:**

I have read and agree with the venue's withdrawal policy on behalf of myself and my co-authors.